# Effects of Typical Antimicrobials on Growth Performance, Morphology and Antimicrobial Residues of Mung Bean Sprouts

**DOI:** 10.3390/antibiotics11060807

**Published:** 2022-06-15

**Authors:** Jing Cao, Yajie Wang, Guanzhao Wang, Pingping Ren, Yongning Wu, Qinghua He

**Affiliations:** 1Department of Food Science and Engineering, College of Chemistry and Environmental Engineering, Shenzhen University, Shenzhen 518060, China; caojing2020@email.szu.edu.cn (J.C.); wangyajie2020@email.szu.edu.cn (Y.W.); wangguanzhao2020@email.szu.edu.cn (G.W.); coolbear001@163.com (P.R.); 2Food Safety Research Unit (2019RU014), Chinese Academy of Medical Science, NHC Key Laboratory of Food Safety Risk Assessment, China National Center for Food Safety Risk Assessment, Beijing 100021, China; wuyongning@cfsa.net.cn; 3Shenzhen Key Laboratory of Food Macromolecules Science and Processing, Shenzhen University, Shenzhen 518060, China

**Keywords:** *Vigna radiata*, mung bean sprouts, antimicrobials, production, morphology, residues

## Abstract

Antimicrobials may be used to inhibit the growth of micro-organisms in the cultivation of mung bean sprouts, but the effects on mung bean sprouts are unclear. In the present study, the growth performance, morphology, antimicrobial effect and antimicrobial residues of mung bean sprouts cultivated in typical antimicrobial solutions were investigated. A screening of antimicrobial residues in thick-bud and rootless mung bean sprouts from local markets showed that the positive ratios of chloramphenicol, enrofloxacin, and furazolidone were 2.78%, 22.22%, and 13.89%, respectively. The cultivating experiment indicated that the production of mung bean sprouts in antimicrobial groups was significantly reduced over 96 h (*p* < 0.05). The bud and root length of mung bean sprouts in enrofloxacin, olaquindox, doxycycline and furazolidone groups were significantly shortened (*p* < 0.05), which cultivated thick-bud and rootless mung bean sprouts similar to the 6-benzyl-adenine group. Furthermore, linear regression analysis showed average optical density of 450 nm in circulating water and average production had no obvious correlation in mung bean sprouts (*p* > 0.05). Antimicrobial residues were found in both mung bean sprouts and circulating water. These novel findings reveal that the antimicrobials could cultivate thick-bud and rootless mung bean sprouts due to their toxicity. This study also proposed a new question regarding the abuse of antimicrobials in fast-growing vegetables, which could be a potential food safety issue.

## 1. Introduction

As a common food in China for more than 2000 years, mung bean (*Vigna radiata*), containing protein and dietary fiber, is known for alleviating heatstroke and reducing swelling in summer [1]. During the germination process of mung beans, biochemical reactions not only produce active compounds including polyphenols, saponins and vitamin C [2], but also decrease phytate content, which can form harmful phytate–protein and phytate–mineral–protein complexes that decrease the bio-availability of essential minerals and affect the use of proteins [3]. Mung bean sprouts also have better nutrition and function than mung beans, e.g., the content and bio-availability of zinc and iron [3]. Previous studies have shown that mung bean sprouts are proven to prevent the increase of total cholesterol, low-density lipoprotein and triglyceride, and to transform inorganic Se compounds into organic Se compounds through bio-transformation, which can improve bio-availability [4]. Moreover, the sprouting of mung beans facilitates the generation of total phenolic compounds to improve anti-oxidant capacity [5,6]. Both the antibacterial activity for meat bacteria and the antiviral activity for the respiratory syncytial and herpes simplex viruses were found in extracts from mung beans and its sprouts [7]. Due to the germination process, mung bean sprouts have high bio-availability, anti-oxidant capacity and antiviral activity. Therefore, mung bean sprouts rich in bioactive compounds can be consumed as functional food.

Although there are many nutritional and functional merits for humans, food safety issues often occur in the cultivating process of mung bean sprouts. The existing evidence demonstrates that mung bean sprouts might be contaminated by *Escherichia coli* and *Salmonella*. For example, the occurrence of *Salmonella* in bean sprouts was reported in Germany and the Netherlands in October and November in 2011 [8], and in England and Northern Ireland from January to March in 2011 [9]. Basically, mung bean sprouts are a suitable medium for food-borne pathogens due to their frequent exposure to “water baths” [10]. Food-borne microbial contamination is likely to occur during the production process, which is likely to result in damage to human health. European regulations stipulate that sprout manufacturers are obliged to detect sprout samples for pathogenic bacteria [11]. In addition to the government’s regulatory requirements, consumers also pay attention to the safety of edible mung bean sprouts. As a common method to remove microbials in the cooking process, washing treatments are proven to be incompletely effective [12]. The growth of micro-organisms may not only cause food hygiene problems, but also impair the production of mung bean sprouts. Therefore, it is important for food safety practices to control the growth of pathogenic bacteria and to supervise food-borne microbial contamination in mung bean sprouts.

A variety of methods have been applied to regulate the quality of mung bean sprouts due to microbial contamination. For example, physical treatments including hot water and ethanol vapors have been evaluated for the quality and storage life of sprouts [13]. Chemical treatment such as biocontrol with endophytic *Bacillus subtilis* [14] has also gained increasing attention in microbial contamination. Combination treatments have good application prospects regarding microbial quality including ultrasound and aqueous chlorine dioxide [15], and plasma-activated water [16], which have an additional effect on seed germination and the seedling growth of mung bean [17]. Although the methods of controlling microbial contamination in mung bean sprouts are becoming more diverse, antimicrobials that inhibit the growth of micro-organisms are still the common antibacterial method in the mung bean sprout industry due to low cost and simple operation. Therefore, antimicrobials are often used to protect the growth of mung bean sprouts in agriculture [18]. In addition, plant growth regulators (PGRs) have been widely used to accelerate plant growth, which can cause food safety risks [19]. For example, two sample surveys showed that the positive rates of 6-benzyl adenine (6-BA) were 1.1% and 2.4% in Gwangju in South Korea. The concentration was 0.01–0.02 mg/kg [20,21]. Similarly, although 6-BA is banned in China, it was still detected in some mung bean sprout samples, which contained 0.11 mg/kg of 6-BA [19]. Even though antimicrobials and PGRs are potentially harmful to human health, they may be still used to enhance the growth performance of mung bean sprouts. Consequently, the contamination of antimicrobials and PGRs has become a problem that needs to be solved.

According to existing research, antimicrobial contamination in mung bean sprouts poses a potential risk to food safety. However, the existing information mainly focuses on the methods for detecting antimicrobials in mung bean sprouts. Few studies have been conducted to research the specific effect of antimicrobials in the cultivation of mung bean sprouts. This study aimed to investigate the role of antimicrobials in the growth of mung bean sprouts, and to clarify antimicrobial residues in the cultivation of mung bean sprouts. First, commercial mung bean sprout samples were collected from local markets and screened by antimicrobial residues. Second, mung beans were cultivated in typical antimicrobial solutions for 96 h using automatic bean sprout machines. Through the experimental cultivation of mung bean sprouts, the effects of typical antimicrobials on production, growth performance and morphology during the growth of mung bean sprouts are explored and the antimicrobial residues in mung bean sprouts are measured.

## 2. Methods

### 2.1. Materials and Reagents

Newly harvested mung bean seeds (*Vigna radiata*) were purchased from a local supermarket in Shenzhen (Guangdong, China). In the market sample survey, a total of 36 samples of short-rooted mung bean sprouts were randomly purchased from local shopping malls, supermarkets, or farmers’ markets in Shenzhen (Guangdong, China). Four samples each were taken from nine districts, namely Futian, Nanshan, Luohu, Baoan, Yantian, Guangming, Longhua, Pingshan, and Dapeng in Shenzhen.

Chloramphenicol, enrofloxacin, olaquindox, doxycycline, furazolidone and 6-BA, formic acid, sodium chloride, ethylenediaminetetraacetic acid (EDTA) and EDTA disodium salt were purchased from Aladdin Industrial Co. Ltd. (Shanghai, China). Chromatographic-grade acetonitrile was obtained from Merck & Co. Inc. (Darmstadt, Germany). All antimicrobial standards were purchased from Sigma–Aldrich (St. Louis, MO, USA). MAS-QuEChERS extraction package was bought from Angela Technologies Co. Ltd. (Tianjin, China). 

### 2.2. Germination and Cultivation of Mung Bean Sprouts

Tap water, distilled water, 5 mg/L 6-BA solution, 100 mg/L chloramphenicol solution, 100 mg/L enrofloxacin solution, 50 mg/L olaquindox solution, 50 mg/L doxycycline solution and 500 mg/L furazolidone solution were added into automatic bean sprout machines (Bear Electric Appliance Co. Ltd., Foshan, China). These antimicrobial solutions were prepared with tap water to simulate the actual situation. After the 24 h of germination-accelerating operation, eight portions of 200 g mung bean seeds were placed into individual automatic bean sprout machines. The setting temperature was 30 °C in the dark environment. The sprout period was 96 h, and all experiments were repeated five times. Each bean sprout machine was fixed in the same treatment to avoid cross-contamination.

### 2.3. Growth Performance Measurement

The sprouts were weighed in dark conditions with closed doors and windows after 24 h, 48 h, 72 h and 96 h, respectively. Total length, bud length and root length were measured after 24 h, 48 h, 72 h and 96 h, respectively. The sprouts were photographed using an iPhone 6 (Apple Inc., Cupertino, CA, USA) with black laboratory benches as the background and a ruler as the standard.

### 2.4. Microbial Reproduction Measurement in Circulating Water

An optical density of 450 nm (OD450) was used to represent the turbidity of circulating water so that it can infer the microbial concentration in the circulating water [22]. During the growth of the mung bean sprouts, 1 mL of the circulating solution was taken every 24 h and was taken to the spectrophotometer to measure the OD450 value of the circulating water.

### 2.5. Sample Collection

At the end of experiment, sprouts were collected and were crushed using a grinder (Bear Electric Appliance Co. Ltd., Foshan, China). The grinder was cleaned completely to avoid cross-contamination. Crushed sprout samples and 10 mL of circulating water were collected with sealed bags and EP tubes, respectively. All samples were stored at −80 °C for inspection.

### 2.6. Sample Extraction and Purification

Sprout samples (5 ± 0.01 g) were weighed into 50 mL centrifuge tubes with a stopper. A total of 9 mL acetonitrile and 1 mL EDTA solution were added into tubes. First, samples were vortexed for 30 s using a vortex mixer (IKA Works GmbH & Co, Germany). Second, the products were put into the ultrasonic cleaner for 15 s and were centrifuged at 8000 rpm for 5 min. Third, 5 mL of supernatant was taken and added into the MAS-QuEChERS extraction package with shaking for 1 min. After centrifugation at 8000 rpm for 5 min, 4 mL of supernatant was taken and was blown to nearly dry with nitrogen using N-EVAP-24 temovap sample concentrator (Organomaition Associates Inc., Berlin, NH, USA). Finally, the above products were diluted to 1 mL with 30% acetonitrile solution and were filtered with a 0.22 μm polyether sulfone filter membrane. In addition, 10 μL circulating water samples were diluted to 10 mL with 30% acetonitrile solution and 1 mL was taken to be filtered with a 0.22 μm polyether sulfone filter membrane. The antimicrobial residues in sprout and circulating water samples were detected using high-performance liquid chromatography-tandem mass spectrometry (HPLC-MS).

### 2.7. Screening of Antimicrobial Residues in Commercial Mung Bean Sprouts

In the market sample survey, antimicrobial residues were screened by a HPLC-Triple Quad^TM^ 4500 mass spectrometer (AB Sciex, Framingham, MA, USA) with an electrospray ionization (ESI) probe. An analytical column (Phenomenex C_18_, 50 × 3.0 mm, 2.6 μm particle size) was employed. Mobile phases were water containing 0.1% formic acid (phase A) and acetonitrile containing 0.1% formic acid (phase B). The flow rate of the mobile phase was 0.4 mL/min, 0.3 mL/min and 0.5 mL/min, which represented the flow rates of positive ion gradient A, positive ion gradient B and negative ion gradient A, respectively. The column temperature was 25 °C and the injection volume was 5 μL. Detailed conditions are listed in Appendix A. Multiple reaction monitoring mode was used to detect antimicrobials in the positive and negative electrospray ionization mode. Nitrogen was employed as a nebulizer and drying gas at 50 psi and 600 °C. The ion spray voltage was set to 5500 V and −4500 V in the positive and negative modes, respectively. Detailed conditions are listed in Appendix A. Concentrations of the antimicrobial residues in commercial mung bean sprouts were calculated.

### 2.8. Detection of Antimicrobial Residues in Sprouts and Circulating Water

The corresponding antimicrobial residues in sprout and circulating water samples collected from chloramphenicol, enrofloxacin, olaquindox, doxycycline and furazolidone groups were determined using an HPLC-Triple Quad^TM^ 4500 mass spectrometer (AB Sciex, Framingham, MA, USA), respectively. The HPLC and MS conditions were as referred to in the method described in the previous section. The concentrations of the antimicrobials and relevant metabolite were calculated.

### 2.9. Statistical Analysis

Data were expressed as mean ± standard deviation (SD). Statistical significance was determined by Student’s *t*-test using the SPSS 23 software (IBM SPSS Statistics, Chicago, IL, USA). The correlation analysis between OD450 values and production was analyzed by Pearson linear regression. A difference where *p* < 0.05 was considered to be statistically significant.

## 3. Results and Discussion

### 3.1. Antimicrobial Screening of Commercial Mung Bean Sprouts

To investigate the status of antimicrobial residues in mung bean sprouts sold in local shopping malls, supermarkets, or farmers’ markets, randomly purchased commercial mung bean sprouts were prospected for antimicrobial residues. Twenty-eight antimicrobials, including chloramphenicol, nitrofuran and quinolones from the sampled bean sprouts, were screened by HPLC-MS [22]. As shown in Table 1, the positive ratios of chloramphenicol, enrofloxacin, and furazolidone were 2.78%, 22.22%, and 13.89% in all samples, respectively.

The occurrence of antimicrobials in commercial mung bean sprouts might be associated with the addition of antimicrobials by manufacturers. It is similar to the antibacterial effect of fungicides in mung bean sprouts. Previous studies have reported that adding fungicides during cultivation can cause harmful residues in the edible parts of plants, which can adversely affect food and the environment [20]. *Salmonella*, *E. coli* and other anaerobic bacteria were the common contaminations affecting the growth of mung bean sprouts. The use of chloramphenicol, enrofloxacin and furazolidone in the cultivation of mung bean sprouts is due to their strong antibacterial properties, which can effectively inhibit the reproduction of bacteria [23,24]. It can greatly increase the economic value of mung bean sprouts. Moreover, antimicrobial pollution of the water used to cultivate mung bean sprouts might be another cause of antimicrobial residue. To save costs, manufacturers use surface water or reclaimed water instead of tap water or purified water in the cultivation process. Surface water for planting might be contaminated by antimicrobials when flowing through soil during agricultural production. After the sewage treatment system, there are still antimicrobial residues in reclaimed water [25]. This can cause plants to be exposed to antimicrobials at a concentration of up to 1 mg/L [26]. Furthermore, the increased bacterial resistance to antimicrobials might lead to an increase in the amount of antimicrobials added during the cultivation process, which can have a harmful impact on food safety [27]. Therefore, antimicrobial abuse and pollution might be the main reasons for antimicrobial residues in commercial mung bean sprouts, which poses a great potential risk to human health. However, the actual effect of antimicrobials on mung bean sprouts is unclear. It is crucial that the cultivated experiments on mung bean sprouts be conducted to clarify the role and residue of antimicrobials in the growth of mung bean sprouts.

### 3.2. Microbial Content in Circulating Water

The inhibitory effects of antimicrobials on micro-organisms were investigated using OD450 values of circulating water. According to Figure 1, the largest OD450 value was found in the tap water group, which was related to the largest turbidity. Turbidity and OD value proved that growth and reproduction of micro-organisms existed in circulating water [28]. OD450 values of circulating water after 96 h in chloramphenicol, enrofloxacin, olaquindox, doxycycline and furazolidone groups were significantly lower than those of tap water, distilled water and 6-BA groups after 96 h (*p* < 0.05). It is suggested that antimicrobials have significant inhibitory effects on the growth of micro-organisms. In agriculture, antimicrobials are used to inhibit the growth of micro-organisms due to the antibacterial effect [18]. In terms of PGRs, the OD450 value of the 6-BA group was not significantly different to that in the tap water group (*p* > 0.05), which indicates that 6-BA had no obvious inhibitory effect on micro-organisms. As mentioned above, microbial growth existed in the cultivation of mung bean sprouts. The growth of micro-organisms is significantly inhibited by the addition of antimicrobials.

### 3.3. Effects of Typical Antimicrobials on Production of Mung Bean Sprouts

The production of mung bean sprouts grown in typical antimicrobial solutions across 96 h is shown in Table 2. The production of mung bean sprouts in all groups increased with growth time across 96 h. Production in the tap water group was the highest at 1012.1 g and production in the furazolidone group was the lowest, at 563.4 g after 96 h. There was no significant difference in the productions of tap water, distilled water and chloramphenicol groups after 96 h (*p* > 0.05). Productions in the 6-BA, enrofloxacin, olaquindox, doxycycline and furazolidone groups were significantly lower than those in the tap water group after 96 h (*p* < 0.05).

Production in the tap water group was higher than that in the distilled water group, which was associated with the presence of Na^+^, Cl^−^, K^−^ and Mg^2+^ in tap water. A previous study proved that Na^+^ and K^−^ facilitated plant growth in agricultural production by participating in the cellular mechanisms [29]. In the antimicrobial groups, the production in the chloramphenicol group was the largest, which might be ascribed to the fact that Cl^-^ could regulate osmotic pressure in plants and cause the guard cells to swell [30]. Production in the furazolidone group was the smallest, which might relate to the toxicity of furazolidone. It was proved that furazolidone affected the absorption of water and soluble nutrients in mung bean sprouts [31].

Interestingly, treatments of antimicrobials showed no improvement on the production of mung bean sprouts compared with tap water. The previous study showed that furazolidone and nifuroxazide containing derivatives of 5-nitrofurfural prevented a fresh matter of oats and radishes from weight gain [32], which was similar to the present experiment. It was suggested that this side effect of antimicrobials might inhibit the proliferation and growth of cells, which reduces the production of mung bean sprouts. Moreover, linear regression results of correlation analysis showed that there is no distinct correlation between average OD450 values and average production across 96 h (*p* > 0.05) (Appendix A). It demonstrates that the concentration of micro-organisms in the circulating water has no obvious relationship with production of mung bean sprouts. These new findings indicate that the use of antimicrobials has no improvement on the production of mung bean sprouts. This might be attributed to the effect of antimicrobials on the types of micro-organisms in circulating water, resulting in reduced production. Previous studies have proved that the bio-diversity of key flora in the soil determines crop production [33]. The influence of specific micro-organisms in circulating water on the production of mung bean sprouts might be cause for further research.

### 3.4. Effects of Antimicrobials on Morphology of Mung Bean Sprouts

The morphology of mung bean sprouts is shown in Figure 2. Long and fine roots are seen on mung bean sprouts in the distilled water and tap water groups from 48 h to 96 h. Extremely short roots or rootless specimens can be seen on mung bean sprouts in the 6-BA group from 48 h to 96 h. Mung bean sprouts in the chloramphenicol and enrofloxacin groups had long roots similar to those of the tap water and distilled water groups. Interestingly, extremely short roots or rootless specimens were also observed on mung bean sprouts in the olaquindox, doxycycline and furazolidone groups, which were similar to rootless mung bean sprouts of the 6-BA group. In addition, thicker and shorter buds were seen on mung bean sprouts in the furazolidone group from 48 h to 96 h, which was similar to those in the 6-BA group from a morphological aspect.

The total lengths, bud lengths and root lengths of the mung bean sprouts are shown in Table 2 and Figure 3. The total length, bud length and root length of mung bean sprouts in the tap water group were the longest of all groups. The total length, bud length and root length of the distilled water and chloramphenicol groups were not significantly different from those of the tap water group (*p* > 0.05). However, the bud lengths of the 6-BA, enrofloxacin, olaquindox, doxycycline and furazolidone groups were significantly shortened (*p* < 0.05). Compared with the tap water group, the root lengths of mung bean sprouts in the enrofloxacin, olaquindox, doxycycline and furazolidone groups were significantly decreased after 96 h (*p* < 0.05). The root lengths of mung bean sprouts in the enrofloxacin, olaquindox, doxycycline and furazolidone groups were not significantly different to the 6-BA group (*p* > 0.05).

It is suggested that the use of antimicrobials affects root length and the morphology of mung bean sprouts. Commercially, a longer root affects appearance and reduces the retail price of mung bean sprouts. Since buds are the edible part of mung bean sprouts, short-rooted or rootless mung bean sprouts are more attractive to consumers. The economic benefit of rootless mung bean sprouts might motivate illegal businesses to abuse 6-BA in China. As a PGR, 6-BA is commonly used to cultivate rootless mung bean sprouts in some countries. However, 6-BA has been banned by the State Administration for Market Regulation of China since 2015 due to its irritation of the esophagus and gastric mucosa [34]. In the present study, 6-BA and furazolidone induced a similar appearance of mung bean sprouts as rootless and thick-bud, which would attract consumers and be sold at a higher price. It is suggested that the reason for the abuse of antimicrobials might be to produce rootless and thick mung bean sprouts to increase illegal profit. Therefore, these new findings demonstrate that antimicrobials might be used by illegal manufacturers to replace 6-BA to cultivate rootless mung bean sprouts.

The significant difference between the chloramphenicol group and the other antimicrobial groups might be due to the presence of Cl^−^. Cl^−^ is an indispensable micro-nutrient for plant growth, which can facilitate elongation growth [35]. Therefore, it might reduce the shortening effect of antimicrobials on the lengths of mung bean sprouts. The lengths of mung bean sprouts in the furazolidone group were the smallest, which might be associated with the toxicity of furazolidone in mung bean sprouts [31]. The bud and root lengths of mung bean sprouts in the enrofloxacin, olaquindox, doxycycline and furazolidone groups were obviously shortened, similar to those in the 6-BA group. This suggests that these antimicrobials exhibit harmful effects on plant growth, which inhibit germ and radicle growth, and shorten the length of bud and root. In addition, the degree of root length shortening in mung bean sprouts in antimicrobial groups was greater than that of bud length shortening. This might be related to the stronger sensitivity of roots to the toxicity of antimicrobials compared to buds during plant growth [36]. These results indicate that antimicrobials can shorten the root and bud length of mung bean sprouts. Significant shortening of root length can bring greater benefits to manufacturers. However, it might also lead to the abuse of antimicrobials and potential food safety risks.

### 3.5. Antimicrobial Residues in Mung Bean Sprouts

Antimicrobial residue in mung bean sprouts and circulating water of antimicrobial groups after 96 h were determined and shown in Table 3. Antimicrobial residues existed in both mung bean sprouts and circulating water in the chloramphenicol, enrofloxacin, olaquindox, doxycycline and furazolidone groups. In addition, a metabolite of furazolidone called 3-amino-2-oxazolidinone (AOZ) was detected in mung bean sprouts and circulating water in the furazolidone group.

Residues of antimicrobial prototypes and metabolite in mung bean sprouts indicate that antimicrobials can be absorbed and metabolized by mung bean sprouts. However, residues of antimicrobial prototypes in circulating water show that antimicrobials cannot be completely absorbed by mung bean sprouts, and a considerable amount of antimicrobials remains in circulating water, causing pollution. The overuse of antimicrobials has caused the occurrence of antimicrobial residue and pollution in plants and soil. Previous studies have shown that furazolidone and nitrofurantoin are efficiently metabolized and produce highly persistent metabolites in plants [37]. Ciprofloxacin and oxytetracycline used in planting vegetables also remain in the soil and the residence time is related to half-life [38]. In addition, AOZ residue in circulating water demonstrates that antimicrobials can be excreted into circulating water after being metabolized by micro-organisms or mung bean sprouts. The metabolic pathway of furazolidone was mainly a reduction of the nitro group in organisms, which might contribute to the generation of reactive metabolites and cause adverse effects [39]. AOZ had a higher stability and longer residence time [40]. Thus, residues of antimicrobials and their metabolites in mung bean sprouts and circulating water might not only cause food safety issues but also induce longer-lasting pollution in surface water and soil.

## 4. Conclusions

Results of this study indicated that antimicrobials not only affect the production and morphology of mung bean sprouts, but also bring on antimicrobial residue. A novel and unexpected observation is that antimicrobials cultivated thick-bud and rootless mung bean sprouts and decreased production, which is a similar effect to 6-BA. Despite the growth inhibition of micro-organisms, antimicrobials cannot increase the production of mung bean sprouts. Additionally, the cultivation study reveals that antimicrobial residues exist in mung bean sprouts and circulating water. Chloramphenicol, enrofloxacin, and furazolidone residue was also found in commercial mung bean sprouts. These findings indicate that the abuse of antimicrobials might increase potential food safety risks in fast-growing vegetables, and the monitoring of antimicrobial residues might be necessary to guarantee food safety.

## Figures and Tables

**Figure 1 antibiotics-11-00807-f001:**
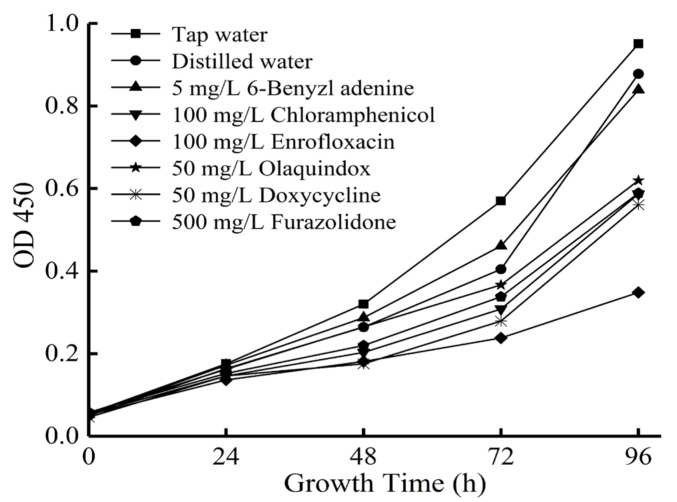
OD450 value of circulating water across 96 h.

**Figure 2 antibiotics-11-00807-f002:**
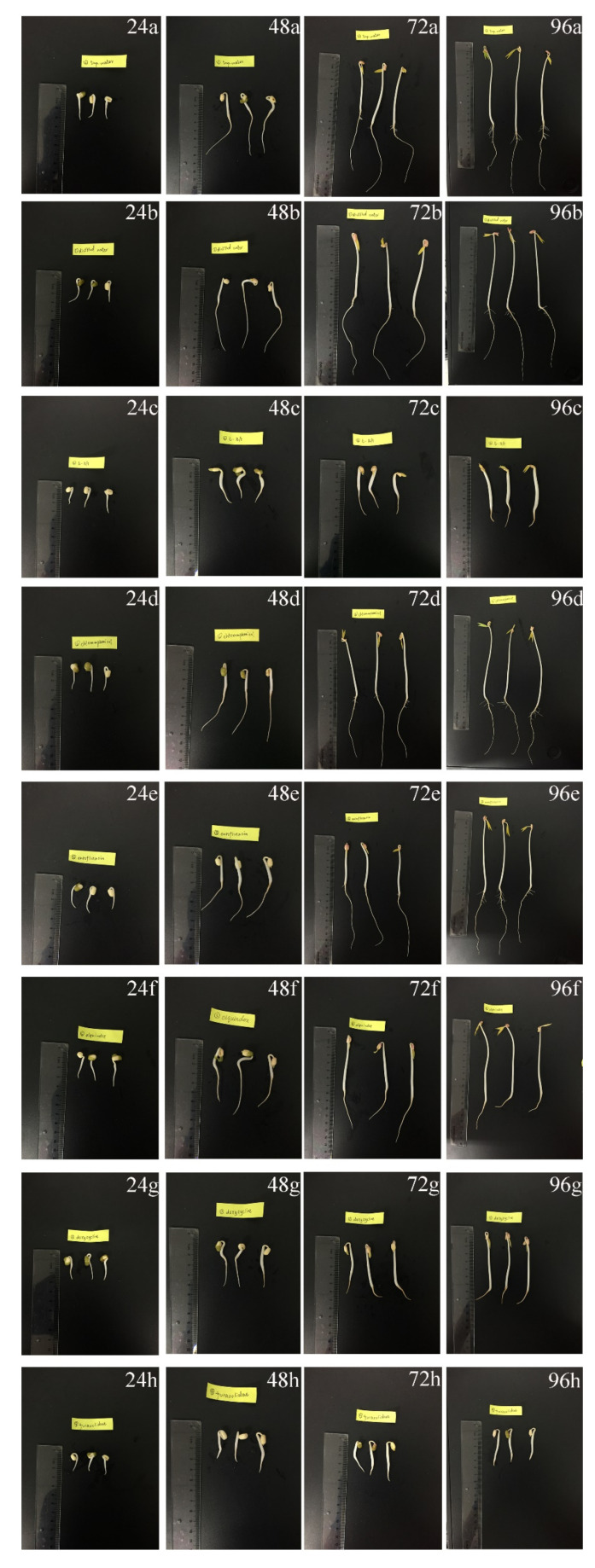
Morphological changes in mung bean sprouts in tap water (**a**), distilled water (**b**), 5 mg/L 6-benzyl adenine (**c**), 100 mg/L chloramphenicol (**d**), 100 mg/L enrofloxacin (**e**), 50 mg/L olaquindox (**f**), 50 mg/L doxycycline (**g**) and 500 mg/L furazolidone (**h**) groups across 96 h. (Group 24, 48, 72 and 96 were the morphologies of mung bean sprouts at 24, 48, 72 and 96 h, respectively. Group (**a**–**h**) represent different antibacterial groups, respectively).

**Figure 3 antibiotics-11-00807-f003:**
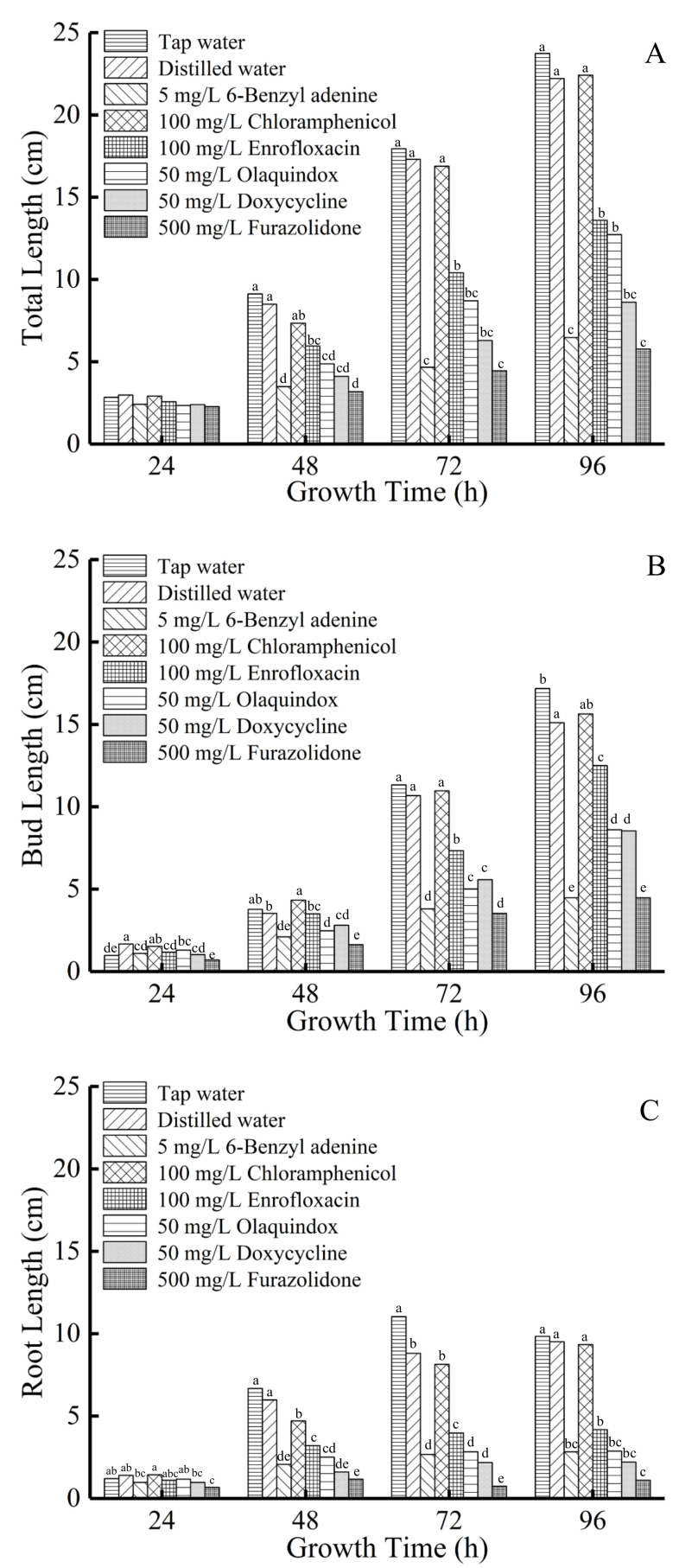
Total length (**A**), bud length (**B**) and root length (**C**) of mung bean sprouts within 96 h. Values with different letters within a column are significantly different (*p* < 0.05).

**Table 1 antibiotics-11-00807-t001:** Concentration (μg/kg) of antimicrobial and metabolite residues in mung bean sprouts from market survey (*n* = 36, mean ± standard deviation).

Residues	Positive Samples	Positive Ratio	Concentration
Chloramphenicol	1	2.78%	9.31
Enrofloxacin	8	22.22%	193.23 ± 98.42
AOZ	5	13.89%	2.88 ± 1.93

**Table 2 antibiotics-11-00807-t002:** Growth performance of mung bean sprouts within 96 h (*n* = 5, mean ± standard deviation).

Groups	Growth Time			
	24 h	48 h	72 h	96 h
Production (g)				
Tap water	382.9 ± 9.2 ^a^	585.2 ± 10.3 ^a^	858.1 ± 60.1 ^a^	1012.1 ± 95.6 ^a^
Distilled water	379.8 ± 16.0 ^a^	567.6 ± 21.7 ^ab^	763.4 ± 115.4 ^abc^	873.9 ± 162.2 ^abcd^
6-Benzyl adenine	371.7 ± 18.9 ^a^	494.9 ± 42.0 ^cde^	629.8 ± 65.3 ^def^	786.5 ± 98.0 ^cde^
Chloramphenicol	381.9 ± 9.4 ^a^	569.4 ± 10.7 ^ab^	811.4 ± 89.5 ^ab^	971.2 ± 142.0 ^abc^
Enrofloxacin	375.0 ± 8.6 ^a^	528.4 ± 19.6 ^bcd^	697.1 ± 77.0 ^bcd^	806.3 ± 143.6 ^bcd^
Olaquindox	348.7 ± 37.0 ^b^	484.1 ± 92.4 ^def^	643.9 ± 199.1 ^cde^	761.9 ± 266.8 ^de^
Doxycycline	360.5 ± 11.6 ^ab^	451.4 ± 20.9 ^ef^	538.9 ± 53.2 ^ef^	595.3 ± 87.4 ^ef^
Furazolidone	359.3 ± 6.9 ^ab^	436.7 ± 24.6 ^f^	512.0 ± 69.1 ^f^	563.4 ± 108.3 ^f^
Total length (cm)				
Tap water	2.83 ± 1.10	9.12 ± 2.23 ^a^	17.94 ± 3.85 ^a^	23.73 ± 1.89 ^a^
Distilled water	2.97 ± 0.28	8.49 ± 1.74 ^a^	17.30 ± 1.39 ^a^	22.21 ± 2.12 ^a^
6-Benzyl adenine	2.41 ± 0.43	3.49 ± 0.68 ^d^	4.66 ± 1.16 ^c^	6.47 ± 1.27 ^c^
Chloramphenicol	2.91 ± 0.54	7.34 ± 1.39 ^ab^	16.88 ± 2.21 ^a^	22.41 ± 1.94 ^a^
Enrofloxacin	2.57 ± 0.35	5.95 ± 0.52 ^bc^	10.40 ± 3.63 ^b^	13.59 ± 5.26 ^b^
Olaquindox	2.34 ± 0.24	4.87 ± 1.13 ^cd^	8.70 ± 5.06 ^bc^	12.72 ± 6.83 ^b^
Doxycycline	2.39 ± 0.18	4.11 ± 1.05 ^cd^	6.29 ± 1.41 ^bc^	8.61 ± 1.94 ^bc^
Furazolidone	2.27 ± 0.52	3.18 ± 0.79 ^d^	4.45 ± 1.81 ^c^	5.77 ± 2.88 ^c^
Bud length (cm)				
Tap water	0.97 ± 0.15 ^de^	3.77 ± 0.21 ^ab^	11.33 ± 0.32 ^a^	17.17 ± 0.95 ^b^
Distilled water	1.67 ± 0.12 ^a^	3.53 ± 0.64 ^b^	10.67 ± 0.45 ^a^	15.10 ± 0.40 ^a^
6-Benzyl adenine	1.10 ± 0.10 ^cd^	2.10 ± 0.10 ^de^	3.80 ± 0.44 ^d^	4.47 ± 0.55 ^e^
Chloramphenicol	1.53 ± 0.15 ^ab^	4.33 ± 0.29 ^a^	10.97 ± 0.32 ^a^	15.63 ± 1.40 ^ab^
Enrofloxacin	1.17 ± 0.15 ^cd^	3.50 ± 0.36 ^bc^	7.33 ± 0.21 ^b^	12.50 ± 0.53 ^c^
Olaquindox	1.30 ± 0.10 ^bc^	2.47 ± 0.25 ^d^	5.00 ± 0.44 ^c^	8.60 ± 0.82 ^d^
Doxycycline	1.03 ± 0.21 ^cd^	2.80 ± 0.44 ^cd^	5.57 ± 0.76 ^c^	8.53 ± 0.21 ^d^
Furazolidone	0.70 ± 0.10 ^e^	1.63 ± 0.32 ^e^	3.53 ± 0.30 ^d^	4.47 ± 0.93 ^e^
Root length (cm)				
Tap water	1.20 ± 0.10 ^ab^	6.67 ± 0.65 ^a^	11.03 ± 0.68 ^a^	9.83 ± 0.55 ^a^
Distilled water	1.40 ± 0.36 ^ab^	5.97 ± 0.50 ^a^	8.80 ± 0.70 ^b^	9.50 ± 2.86 ^a^
6-Benzyl adenine	0.97 ± 0.06 ^bc^	2.07 ± 0.25 ^de^	2.67 ± 0.21 ^d^	2.83 ± 0.25 ^bc^
Chloramphenicol	1.43 ± 0.15 ^a^	4.70 ± 0.78 ^b^	8.13 ± 0.85 ^b^	9.33 ± 0.67 ^a^
Enrofloxacin	1.10 ± 0.10 ^abc^	3.20 ± 0.66 ^c^	3.97 ± 0.67 ^c^	4.17 ± 0.95 ^b^
Olaquindox	1.17 ± 0.38 ^ab^	2.50 ± 0.20 ^cd^	2.83 ± 0.35 ^d^	2.87 ± 0.29 ^bc^
Doxycycline	0.97 ± 0.25 ^bc^	1.60 ± 0.36 ^de^	2.17 ± 0.40 ^d^	2.20 ± 0.36 ^bc^
Furazolidone	0.67 ± 0.12 ^c^	1.17 ± 0.21 ^e^	0.73 ± 0.06 ^e^	1.10 ± 0.20 ^c^

Values with different letters within a column are significantly different (*p* < 0.05).

**Table 3 antibiotics-11-00807-t003:** Concentration (μg/kg) of antimicrobial and metabolite residues in mung bean sprouts and circulating water from cultivating experiment (*n* = 5, mean ± standard deviation).

Groups	Antimicrobial Residues	Mung Bean Sprouts	Circulating Water
Chloramphenicol	Chloramphenicol	45.6 ± 4.8	82.9 ± 6.9
Enrofloxacin	Enrofloxacin	93.1 ± 2.8	84.2 ± 2.9
Olaquindox	Olaquindox	17.6 ± 4.9	41.6 ± 2.9
Doxycycline	Doxycycline	37.7 ± 2.0	44.1 ± 3.0
Furazolidone	Furazolidone	388.6 ± 18.1	458.9 ± 29.9
	AOZ	24.1 ± 3.6	19.3 ± 3.7

## Data Availability

All data generated or analyzed during the study appear in the submitted article.

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
