# Peer review of "Effects of Typical Antimicrobials on Growth Performance, Morphology and Antimicrobial Residues of Mung Bean Sprouts"

_antibiotics, 2022, doi:10.3390/antibiotics11060807_

Round 1

Reviewer 1 Report

The given topic about the effect of antibiotics on the mung sprouts features is very interesting and highly important for food safety. The article is written in a very good form in all neccessary part. The only minor formal error is that the lines names miss in Figure 2. Generally, I recommend this article for publishing in the present version. 

Reviewer 2 Report

Comments and suggestions for authors are provided below and for supplementary materials in an attached file.

Journal: Antibiotics

Review comments for manuscript entitled:

Effects of Typical Antimicrobials on Growth Performance, Morphology and Antimicrobial Residues of Mung Bean Sprouts

  1. Line 43: “…meat bacterial…” change to “…meat bacteria…”
  2. Line 78: “…risks.[19]” change to “…risks [19].”
  3. Line 123: “…were repeated for…” delete “for” and change to “…were repeated…”
  4. Line 154: “…and 1 mL were taken…” change to “…and 1 mL was taken…”
  5. Line 169: “…employed as nebulizer…” add “a” and change to “…employed as a nebulizer…”
  6. Line 191: “…were detected for…” change to “were prospected for”
  7. Table 1. and Line 198 : there are no letters within a column in the table which define the statistical significances
  8. Line 242: “…was shown…” change to “…is shown…”
  9. Line 293: “…were shown…” change to “…are shown…”
  10. Line 297: “…(P <05)…” change to “…(P > 0.05)…”
  11. Line 337: “Antimicrobial residues were…” delete “were” and change to “Antimicrobial residues…”
  12. Line 348: “Over use…” change to “Overuse…”
  13. Line 349: “…caused residue and pollution of antimicrobials in…” change to “…caused occurrence of antimicrobial residues and pollution in…”

Reviewer 3 Report

Reviewer A

Comment’s:

I have completed a review of the manuscript “Effects of typical antimicrobials on growth performance, morphology and antimicrobial residues of mung bean sprouts” (Antibiotics - 1738611). The aim of this study was to understand and elucidate This study aimed to investigate the role of antimicrobials in the growth of mung bean sprouts and to clarify antimicrobial residues in the cultivation of mung bean sprouts. I congratulate the authors for their excellent work, the research is of great importance for the field of the pharmacology and has technical and scientific merit. The information generated will be of great value. The findings indicated that the abuse of antimicrobials might cause potential food safety risks in fast growing vegetables and the monitoring of antimicrobial residues might be necessary to guarantee food safety. Overall the manuscript is a well-designed study, the figures are of good quality and the text is mostly adequate, data were analyzed properly, the search results are consistent and very interesting and it fits the scope of the journal well.
